# The Incidence of Chronic Limb-Threatening Ischemia in the Midland Region of New Zealand over a 12-Year Period

**DOI:** 10.3390/jcm11123303

**Published:** 2022-06-09

**Authors:** Odette Hart, Nicole Xue, Brittany Davis-Havill, Mark Pottier, Minesh Prakash, Sascha-Akito Reimann, Jasmin King, William Xu, Manar Khashram

**Affiliations:** 1Faculty of Medical and Health Sciences, The University of Auckland, Auckland 1010, New Zealand; nxue896@aucklanduni.ac.nz (N.X.); bdav518@aucklanduni.ac.nz (B.D.-H.); mpot950@aucklanduni.ac.nz (M.P.); wxu958@aucklanduni.ac.nz (W.X.); manar.khashram@gmail.com (M.K.); 2Department of Vascular and Endovascular Surgery, Waikato District Health Board, Hamilton 3204, New Zealand; minesh.prakash@waikatodhb.health.nz (M.P.); sascha-akito.reimann@waikatodhb.health.nz (S.-A.R.); jasmin.king@waikatodhb.health.nz (J.K.)

**Keywords:** epidemiology, incidence, peripheral artery disease, arteriosclerosis, intermittent claudication, chronic limb-threatening ischemia, health services research

## Abstract

The epidemiology of severe PAD, as characterized by short-distance intermittent claudication (IC) and chronic limb-threatening ischemia (CLTI), remains undefined in New Zealand (NZ). This was a retrospective observational cohort study of the Midland region in NZ, including all lower limb PAD-related surgical and percutaneous interventions between the 1st of January 2010 and the 31st of December 2021. Overall, 2541 patients were included. The mean annual incidence of short-distance IC was 15.8 per 100,000, and of CLTI was 36.2 per 100,000 population. The annual incidence of both conditions was greater in men. Women presented 3 years older with PAD (*p* < 0.001). Patients with short-distance IC had lower ipsilateral major limb amputation at 30 days compared to CLTI (IC 2, 0.3% vs. CLTI 298, 16.7%, *p* < 0.001). The 30-day mortality was greater in elderly patients (<65 years 2.7% vs. ≥65 years 4.4%, *p* = 0.049), but did not differ depending on sex (females 36, 3.7% vs. males 64, 4.1%, *p* = 0.787). Elderly age was associated with a worse survival for both short-distance IC and CLTI. There was a worse survival for females with CLTI. In conclusion, PAD imposes a significant burden in NZ, and further research is required in order to reduce this disparity.

## 1. Introduction

Peripheral artery disease (PAD) was estimated to affect over 236 million people globally in 2015 [1], increasing from 202 million in 2010 [2]. PAD is associated with a 10–15 times greater risk of cardiovascular (CV) mortality and a 2–5-fold increase in all-cause mortality compared to those without PAD [3,4,5,6,7]. The mortality risk appears to be greater for men than women [5,8]. 

While a large number of individuals with PAD are asymptomatic or experience stable, mild lower limb symptoms from their atherosclerotic disease, a portion will progress to severely symptomatic disease which might require surgical and percutaneous intervention [9,10]. These cases are observed as short-distance intermittent claudication (IC) and chronic limb-threatening ischemia (CLTI). CLTI is the experience of lower limb rest pain, chronic ulceration and gangrene, and probably represents around 10% of all PAD [9,11,12]. The annual incidence of CLTI has previously been estimated at 500–1000 new cases per million people in Western countries [9,10].

The management of short-distance IC and CLTI warrants specialist vascular input, multidisciplinary care, and may involve complex wound care and frequent readmission to hospital [9,13]. These factors culminate in significant patient burden and health system impact. Thus, understanding the incidence of these severe disease states of PAD, and any discrepancies among subgroups, is important for health care planning and fair resource management. Yet within the NZ population, there is a lack of data on PAD epidemiology, clinical presentation and outcomes.

Establishing disease incidence via prospective methods is challenging, secondary to the difficulties of obtaining a large enough sample size that is sufficiently distributed both sociologically and geographically to represent the general population [14]. Prospective incidence studies in PAD have used a change in ankle-brachial index (ABI) value of <0.9 during a defined follow-up period as a measure of PAD incidence [7]; however, these studies are limited by recruitment numbers, biases in sample selection and short follow-up timeframes. Furthermore, from these studies it can be difficult to ascertain the incidence of short-distance IC and CLTI from less severe forms of PAD which do not require intervention. Large administrative databases are often utilised to obtain valid and reliable estimates of disease incidence [14,15]; when applied to PAD and combined with extensive review of each individuals history to collect patient level data, such information can help form an understanding of the current patient and health system impact from the management of severely symptomatic PAD.

Therefore, the aim of this study was to define the incidence of surgical and percutaneous-treated short-distance IC and CLTI, and report the outcomes for the Midland region of NZ.

## 2. Materials and Methods

**Study design:** Reporting of this observational study followed the Strengthening the Reporting of Observational Studies in Epidemiology (STROBE) guidelines [16]. 

This was a retrospective observational cohort study of patients in the Midland region of NZ that underwent surgical and endovascular intervention for PAD. The Midland region has 19.9% (1,024,860) of the NZ total population in 2022, and covers 21% (56,728 km^2^) of NZ’s land mass [17,18]. All patients that underwent surgical and endovascular intervention for lower limb short-distance IC and CLTI were included from the tertiary referral center (Waikato Hospital, NZ) between 1 January 2010 and 31 December 2021. PAD treatments included were endovascular revascularization, open surgical revascularization, minor limb amputation and major limb amputation (MLA). The exclusion criteria were the following:Age < 18 years.Surgical or endovascular management of acute lower limb ischemia.Surgical or endovascular management of aneurysmal disease (unless PAD symptomology with angiographic evidence of PAD concomitant).Diagnostic lower limb angiogram without angioplasty.Repeated revascularization of previous bypass or endovascular intervention.Amputation for trauma, tumor, acute limb ischemia or diabetic foot disease without PAD or infection.

Ethics approval for this study was granted by New Zealand Central Health and Disability Ethics Committee (20/CEN/122) and the Waikato District Health Board Research Department (RDO020044).

**Study protocol and data collection:** The primary outcome was the incidence of surgical and percutaneous treated IC and CLTI within the Midland region. Secondary outcomes were comparison of 30-day MLA rate, amputation-free survival, 30-day mortality and overall survival between sex. 

Patients that underwent PAD management were identified from hospital-based operation reports at Waikato Hospital and the Australasian Vascular Audit (AVA). Manual review of each electronic medical record was conducted in order to ensure Midland region residency; review of each patient’s clinical notes and arterial imaging was conducted in order to confirm that short-distance IC and CLTI were the indication for surgery. Demographic and surgical procedure information was gathered directly from hospital medical records; this included age at index operation, sex, and the presence of diabetes mellitus (DM) defined as hemoglobin A1c greater than 48mmol/mol; end stage renal failure (ESRF) defined as a decline in kidney function to a level at which either dialysis or kidney transplantation is required; chronic obstructive pulmonary disease (COPD) defined as progressive pulmonary disease with not fully reversible airflow limitation; ischemic heart disease (IHD) defined as history of cardiac symptoms secondary to coronary stenosis; congestive heart failure (CHF) defined as cardiac ejection fraction ≤ 40%; current or previous cancer; diagnosed mild cognitive impairment (MCI) defined as a cognitive deterioration with preserved activities of daily life and intact/minimally impaired complex instrumental functions; and diagnosed dementia defined as a decline in cognitive function of major cognitive domains which interferes with independence in everyday activities. Index surgical procedure information including the date of the index operation and the indication for surgery were collected. Outcomes of MLA and date of death were recorded.

Each treated limb was considered as a separate entity for entry into the database. The first limb procedure (may have been either revascularization or amputation) was considered the index operation for the limb. The nation-wide patient unique identifier (National Health Index, NHI) was utilized to ensure that each patient was entered into the database once only for incidence calculations. Short-distance IC was defined as cramping pain or discomfort in the lower limbs that is reproducibly induced by exercise and relieved by rest [19], and occurring at a short distance that affects quality of life. CLTI was defined as the “presence of PAD in combination with rest pain, gangrene or a lower limb ulceration > 2 weeks duration” [9].

**Statistical analysis**: Data were collected in Excel software (Microsoft, Redmond, Wash), and statistical analysis was performed using R (R Foundation for Statistical Computing, Vienna, Austria). Incident index limb operations were considered only if the patient was a Midland region resident on the procedure date. For patients with bilateral PAD, only the earliest index limb procedure of either leg was used as the incident event. Incidence rates were calculated per 100,000 people. Annual age-specific, sex-specific and ethnicity-specific population counts for the Midland region were obtained from 2013 and 2018 census data of the usual resident population count, retrieved from the national census database [17]. These figures were used as the denominator to calculate the incidence rates for each year. The resident population was standardized according to ethnicity and age, with two age minimums used. Age of >45 years was used for non-Māori patients [2,6], and age of >35 used for NZ Māori patients given the earlier age of onset of CVD in the NZ Māori population [20]. Continuous data are summarized as median (IQR) or as mean (SD) where appropriate, and categorical data are summarized as percentages. Additional analyses were conducted separately for men and women. The 95% confidence intervals (CI) for the incidence estimates were calculated assuming the cases follow a Poisson distribution. The 30-day MLA and mortality rates were calculated. Survival analysis was conducted using Kaplan–Meier survival curves and groups were compared with the log-rank statistic. A *p* value < 0.05 was considered significant.

## 3. Results

### 3.1. Demographics

A total of 2541 patients were included in the study, and 37.8% were female (Table 1). Overall, 43.4% of the cohort had diabetes at the first presentation. The median age of patients presenting with short-distance IC was 71 (63–77) years and CLTI was 72 (62–81) years. Women presented 3 years older with short-distance IC (*p* = 0.024) and CLTI (*p* < 0.001) compared to men.

Overall, 30.2% of patients presented with short-distance IC. More men with short-distance IC experienced concurrent COPD compared to women with short-distance IC (*p* = 0.02).

Patients with CLTI made up 69.8% of the cohort. Men with CLTI experienced more diabetes (*p* = 0.002) and IHD (*p* = 0.016) compared to females with CLTI. The patient demographics are summarized in Table 1.

### 3.2. Incidence

Table 2 shows the age and sex-standardized annual incidences of surgical and percutenous-treated short-distance IC and CLTI within the Midland region. There was a stable annual number of incident cases of PAD-related surgery (combined short-distance IC and CLTI) over the 12 years, with the number of yearly new cases ranging from 184–233.

There was general decrease in the annual incidence of short-distance IC and CLTI over the 12 years. The overall annual incidence of CLTI was higher than short-distance IC.

The male annual incidence of both short-distance IC and CLTI was higher than the female annual incidence (Figure 1). Despite the general decline in annual incidence of CLTI for males, there appears to have been sharp spike in male presentation with CLTI during the 2020–2021 time period.

### 3.3. Impact of Age

The distribution of age at first presentation of females was skewed left for both short-distance IC and CLTI (Figure 2). For both male and female cohorts, a greater portion of patients presented for the first time ≥65 years (female IC 73.2%, CLTI 70%; and male IC 70.8%, CLTI 70.1%) compared to those aged <65 years.

There appears to be a decrease in the age- and sex-standardized median age for female and males at first PAD-related limb procedure from 2010 to 2021; however, there is considerable overlap of IQR, hence no trend is discernable (Figure 3).

### 3.4. Outcomes

#### 3.4.1. Major Limb Amputation

Overall, 298 patients (11.7%) underwent an MLA of the ipsilateral limb within 30 days of their first surgical and percutaneous intervention for PAD.

For patients undergoing surgical and percutaneous intervention for PAD, there was no difference in 30-day ipsilateral MLA for those aged <65 years or ≥65 years (<65 years 99, 13.3% vs. 199, 11.1% in ≥65 years, *p* = 0.123). There was also no difference in ipsilateral MLA at 30 days between sex for patients treated with surgical and percutaneous intervention for PAD (female 110, 11.4% vs. male 188, 11.7%, *p* = 0.779).

However, patients that presented with short-distance IC had a lower percentage of ipsilateral MLA at 30 days compared to CLTI (IC 2, 0.3% vs. CLTI 298, 16.7%, *p* < 0.001).

#### 3.4.2. Survival

The overall 30-day mortality rate for surgical and percutaneous treated PAD was 3.9% (100). A difference in the 30-day mortality was seen for those aged <65 years compared to ≥65 years (< 65 years 2.7% vs. ≥65 years 4.4%, *p* = 0.049). However, the 30-day mortality did not differ between sex (females 36, 3.7% vs. males 64, 4.1%, *p* = 0.787).

There was a statistical difference in the 30-day mortality for short-distance IC compared to CLTI (IC 8, 1.0% vs. CLTI 92, 5.2%, *p* < 0.001). After exclusion of 30-day mortality, the median survival for patients with short-distance IC was 10.4 years compared to 4.2 years for patients with CLTI.

Figure 4a shows that there was no difference in the survival between male and female patients that underwent surgical and percutaneous treated PAD (*p* = 0.24). However, patients that underwent surgical and percutaneous treated PAD with an age <65 years on their first presentation had better survival compared to those treated for the first time after 65 years old (*p* < 0.0001; Figure 4b).

The survival post incident limb surgery was better for patients that presented with short-distance IC compared to CLTI (*p* < 0.0001; Figure 4c).

For short-distance IC, there was no difference in survival between sexes (*p* = 0.16; Figure 4d), yet there was for age stratification (*p* < 0.0001; Figure 4e).

For CLTI, men (*p* = 0.019; Figure 4f) and those aged <65 years (<0.0001; Figure 4g) had better survival.

## 4. Discussion

The mean annual incidence of surgical and percutaneous-treated short-distance IC was 15.8 per 100,000 population and CLTI was 36.2 per 100,000 population within the Midland region. The incidence of short-distance IC and CLTI generally appears to decrease over the 12 years. The annual incidence of both conditions was greater for men compared to women.

Patients with CLTI underwent more MLA in the first 30 days compared to those with short-distance IC. Sex and elderly age (≥65 years) were not associated with 30-day MLA.

Patients with CLTI had a higher 30-day mortality compared to short-distance IC. Elderly age (≥65 years) was associated with higher 30-day mortality and reduced survival after surgical and percutaneous PAD intervention. Sex was not associated with 30-day mortality; however, men had better long-term survival with CLTI compared to women.

The Midland region incidence cannot be compared on a national level given the lack of PAD epidemiology. International studies report higher IC incidences than what we found in this study [21,22,23]. The Framingham Study reported an age- and sex- adjusted incidence of IC of 225/100,000 in the 1990s, which is now likely to be lower given the 18% decline over the prior decade [24]. These studies, however, often base IC off standardized questionnaires and ABI < 0.9, hence likely include long-distance IC and asymptomatic PAD patients, which our study does not.

In this study, the incidence of CLTI is lower than global estimates at 50–100 per 100,000 in Western countries [9,10]. Moreover, a retrospective study of a large insured population in the United States assessing claims in an ≥40 year-old population reported a mean annual incidence of primary CLTI of 190 per 100,000 population [11].

PAD prevalence (as defined as ABI < 0.9) in high-income countries is widely reported to be increasing [1,2,12]. This may be to the result of an aging population [25] in addition to capture of asymptomatic or mildly symptomatic patients in these estimates. Despite this, the age-standardized incidence of symptomatic PAD requiring medical or invasive intervention appears to have decreased overtime [21,24], as occurred in this study. In the United Kingdom, the incidence of symptomatic PAD decreased steadily over time from the year 2000 (385.5/100,000) to 2014 (173.3/100,000) [25], while another recent UK study reported a 15% decline in PAD incidence from 2006 to 2015 [26]. The decline in symptomatic PAD incidence may be the result of an increased uptake of secondary CV prevention strategies, greater use of CV medications and declining smoking rates [25].

The incidence of PAD appears largely affected by the demographics of the observed cohort, including ethnicity, age, sex, and presence of comorbidities [9,27,28,29]. This study used patient level data, and this approach differs with large administrative databases or insurance claim data. The clinical and hemodynamic information required to diagnose short-distance IC and CLTI is not obtainable from large databases lacking detailed patient information [9]. This study’s accuracy was increased by the individual review of each patient’s history, including ensuring diagnostic evidence of PAD prior to inclusion. Additionally, incidence overestimation may occur in studies that use MLA as a surrogate for PAD, given that >80% of MLAs are secondary to CLTI [9]. This study excluded MLA due to trauma, tumor, acute limb ischemia, diabetes without PAD, and infection.

There are well-documented differences in PAD epidemiology and symptomology between sexes. Male sex has long been associated with higher rates of PAD; however, globally the prevalence appears higher in women [30]. Women present more often with rest pain and atypical leg symptoms [29], whereas men present with more severe disease demonstrated by lower ABI, IC and CLTI symptomology [7,31]. Our study suggests that males within the Midland region present either more typically or severely with PAD to initiate surgical intervention. Women were slightly older when undergoing their first PAD-related procedure, and a recent metanalysis reflects this, reporting that females were 2.25 years older at presentation [29]. However, studies report that male sex is associated with a 13% relative increase in all-cause mortality for patients with PAD [8]. Interestingly in our study, long term survival was better for men with CLTI. 

Lower limb PAD increases with age, generally appearing in persons over 50 years, with a rapid increase after 65 years [7]. In the elderly, the gender predisposition towards men becomes mitigated, with women and men presenting more equally [32]. In our study, the age at presentation was around 70 years, and this is typical of disease experience seen elsewhere [31]. For both sexes, around 70% of patients were over 65 years at the time of their first procedure. Furthermore, elderly age (≥65 years) was associated with a higher 30-day mortality and reduced survival. In NZ, the older population is increasing faster than the overall population; hence, the percentage of elderly residents will rise in years to come [33]. Thus, this knowledge of worse PAD outcomes for an elderly population increasing in number is important for clinical decision making and NZ service providers.

The NZ annual incidence of MLA was suggested to range from 6.1 to 7.6 per 100,000 inhabitants during the years 2010 to 2014 [34]. In our study, the presence of CLTI resulted in more MLA at 30 days compared to IC. CLTI is a known risk factor for MLA, and Australian data suggests that patients with CLTI undergoing revascularization have a four-fold increased risk of amputation at two years compared with patients revascularized without CLTI [35].

Lower limb PAD is a marker of generalized atherosclerosis, increased risk of CV events and mortality [32]. This risk increases with PAD severity, hence there is a three-fold increase in major CV events for patients with CLTI compared to IC [32]. Furthermore, CLTI carries a one-year mortality of 20–25%, while IC is 5.7% [31]. In our study, the higher 30-day mortality and worse survival for patients with CLTI concur with this.

A strength of this study was the collection and accurate assessment of patient level data collection which occurred over a relatively long study period. Additionally, our study did not require patient participation in a longitudinal format, hence abating the risk of loss of patients to follow up.

However, this study has limitations. Firstly, data were collected retrospectively. In order to mitigate potential documentation errors, patient level data were collected from multiple hospital records and imaging sources. The selection criteria for offering PAD interventions are based on real-world data, not systematic. There is a cohort of patients with short-distance IC and CLTI that underwent non-operative management that have not been captured. This cohort did not include patients managed solely by general practitioners, private vascular surgeons (however, this represents a small number of PAD surgeries in the Midland region), other surgical specialities, patients that underwent vascular treatment in small regional hospitals or hospitals outside the Midland region, and patients that did not seek medical help.

Secondly, claudication results cannot be assumed to define the experience of long-distance IC, which is treated non-operatively at our centre. This study does not provide epidemiology for the entire spectrum of PAD, as patients with asymptomatic PAD were not included. Future studies could aim to establish the epidemiology of asymptomatic PAD and conservatively managed IC and CLTI within NZ.

Thirdly, the time of patient capture does not coincide with actual appearance of disease. Incidence calculations were based on the date of first PAD surgery or percutaneous intervention. This may cause an artificial extension, as the atherosclerosis and symptomology may have developed earlier.

Larger and future prospective studies are required in this region, in order to improve the management and outcomes of patients with PAD in New Zealand [36].

## 5. Conclusions

In conclusion, PAD imposes a significant burden in NZ. The incidence and outcomes appear to be affected by sex and age. Further national and international research is required in the field of PAD epidemiology.

## Figures and Tables

**Figure 1 jcm-11-03303-f001:**
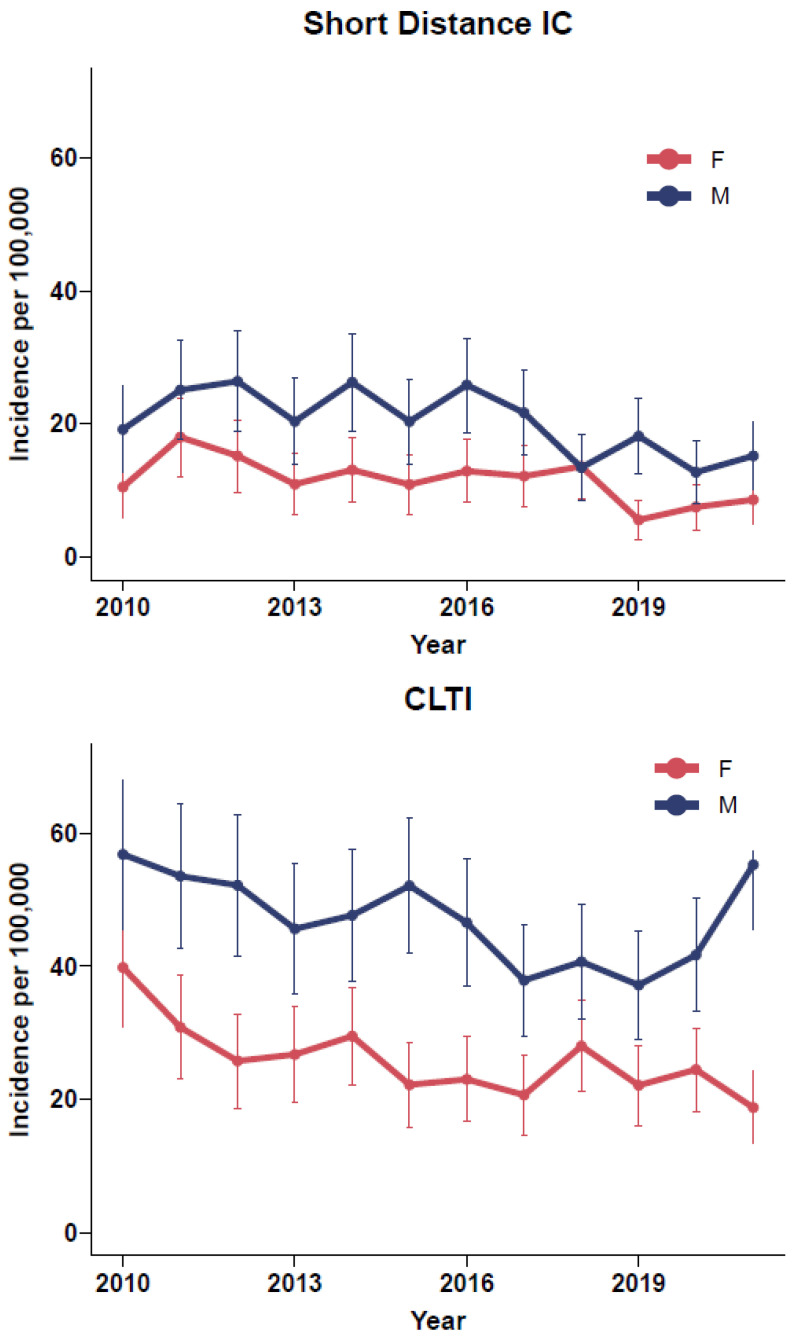
The annual incidence of surgical and percutaneous treated IC and CLTI stratified by sex.

**Figure 2 jcm-11-03303-f002:**
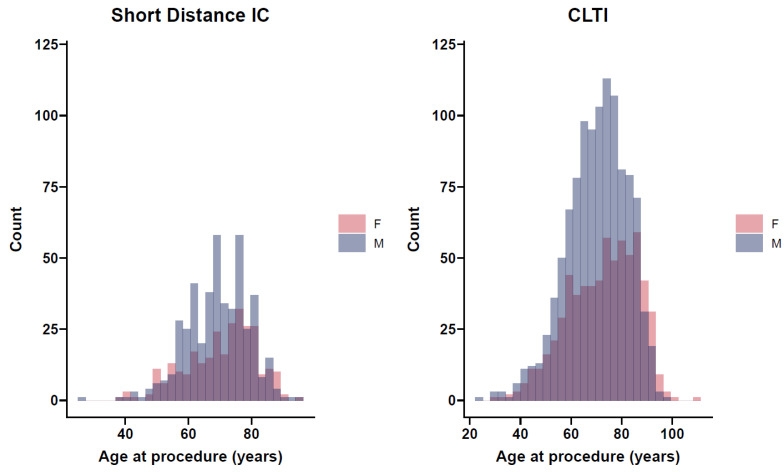
Histogram of age at first presentation of short-distance IC and CLTI as stratified by sex.

**Figure 3 jcm-11-03303-f003:**
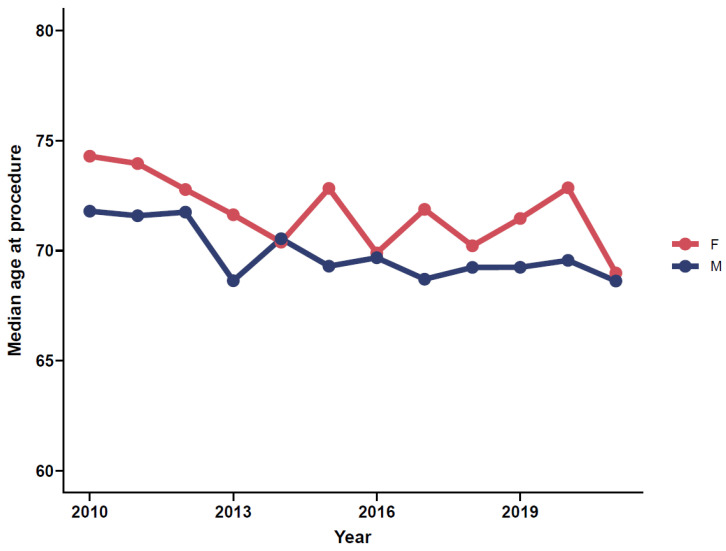
Median age of patients at first surgical and percutaneous treated PAD intervention (short-distance IC and CLTI combined), as separated by sex.

**Figure 4 jcm-11-03303-f004:**
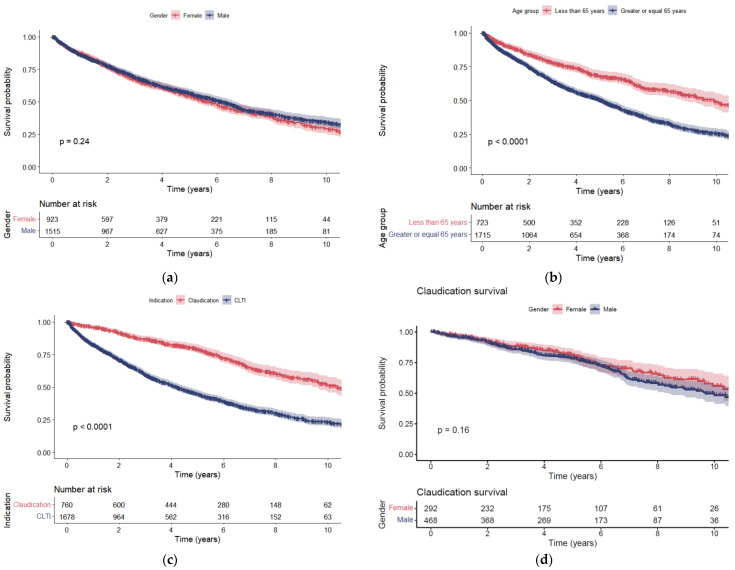
Kaplan–Meier survival curves for (**a**) sex stratification of patients with PAD (combined short-distance IC and CLTI cohorts), (**b**) age stratification of patients with PAD (combined short-distance IC and CLTI cohorts), (**c**) short-distance IC compared to CLTI cohort, (**d**) sex stratification of patients with short-distance IC, (**e**) age stratification of patients with short-distance IC, (**f**) sex stratification of patients with CLTI, and (**g**) age stratification of patients with CLTI.

**Table 1 jcm-11-03303-t001:** Demographics of all patients at the time of first presentation with short-distance IC and CLTI.

		Short-Distance IC				CLTI			
Variable		Female	Male	Overall	*p* Value	Female	Male	Overall	*p* Value
N (%)		295 (38.4)	473 (61.6)	768		666 (37.6)	1107 (62.4)	1773	
Age, median (IQR)		73 (64–79)	70 (62–77)	71 (63–77)	**0.024**	74 (62–83)	71 (63–79)	72 (62–81)	**<0.001**
DM (%)	DM	66 (22.4)	114 (24.1)	180 (23.4)	0.234	313 (47.0)	610 (55.1)	923 (52.1)	**0.002**
	Pre-DM	33 (11.2)	36 (7.6)	69 (9.0)		29 (4.4)	53 (4.8)	82 (4.6)	
ESRF (%)		6 (2.0)	7 (1.5)	13 (1.7)	0.771	75 (11.3)	134 (12.1)	209 (11.8)	0.647
COPD (%)		52 (17.6)	54 (11.4)	106 (13.8)	**0.02**	105 (15.8)	145 (13.1)	250 (14.1)	0.136
IHD (%)		99 (33.6)	183 (38.7)	282 (36.7)	0.175	212 (31.8)	416 (37.6)	628 (35.4)	**0.016**
CHF (%)		21 (7.1)	30 (6.3)	51 (6.6)	0.786	122 (18.3)	211 (19.1)	333 (18.8)	0.745
Cancer (%)		45 (15.3)	68 (14.4)	113 (14.7)	0.819	113 (17.0)	168 (15.2)	281 (15.8)	0.351
Cognitive Impairment (%)	Dementia	1 (0.3)	3 (0.6)	4 (0.5)	0.736	626 (94.0)	1050 (94.9)	1676 (94.5)	0.403
	MCI	3 (1.0)	7 (1.5)	10 (1.3)		32 (4.8)	40 (3.6)	72 (4.1)	
	None	291 (98.6)	463 (97.9)	754 (98.2)		626 (94.0)	1050 (94.9)	1676 (94.5)	

IQR, interquartile range; DM, diabetes mellitus; ESRF, end stage renal failure; COPD, chronic obstructive pulmonary disease; IHD, ischemic heart disease; CHF, congestive heart failure; MCI, mild cognitive impairment.

**Table 2 jcm-11-03303-t002:** The incidence of short-distance IC and CLTI in Midland region as separated by sex. Shown are 95% confidence intervals.

		Short-Distance IC						CLTI					
	Midland	Female ^2^		Male ^2^		Overall		Female ^2^		Male ^2^		Overall	
Year	Pop ^1^	N	Incidence	N	Incidence	N	Incidence	N	Incidence	N	Incidence	N	Incidence
2010	363,372.5	20	10.5 (5.9–15.1)	33	19.1 (12.6–25.7)	53	14.6 (10.7–18.5)	76	39.8 (30.9–48.8)	98	56.8 (45.6–68.1)	174	47.9 (40.8–55)
2011	370,230	35	18 (12–23.9)	44	25.1 (17.7–32.5)	79	21.3 (16.6–26)	60	30.8 (23–38.6)	94	53.6 (42.7–64.4)	154	41.6 (35–48.2)
2012	376,277.5	30	15.1 (9.7–20.6)	47	26.4 (18.8–33.9)	77	20.5 (15.9–25)	51	25.8 (18.7–32.8)	93	52.2 (41.6–62.8)	144	38.3 (32–44.5)
2013	383,995	22	10.9 (6.3–15.4)	37	20.3 (13.8–26.9)	59	15.4 (11.4–19.3)	54	26.7 (19.6–33.9)	83	45.6 (35.8–55.4)	137	35.7 (29.7–41.7)
2014	393,667.5	27	13 (8.1–18)	49	26.2 (18.9–33.6)	76	19.3 (15–23.6)	61	29.5 (22.1–36.9)	89	47.7 (37.8–57.6)	150	38.1 (32–44.2)
2015	403,932.5	23	10.8 (6.4–15.3)	39	20.3 (13.9–26.7)	62	15.3 (11.5–19.2)	47	22.2 (15.8–28.5)	100	52.1 (41.9–62.3)	147	36.4 (30.5–42.3)
2016	415,117.5	28	12.9 (8.1–17.6)	51	25.8 (18.7–32.9)	79	19 (14.8–23.2)	50	23 (16.6–29.4)	92	46.6 (37.1–56.1)	142	34.2 (28.6–39.8)
2017	426,322.5	27	12.1 (7.5–16.7)	44	21.6 (15.2–28)	71	16.7 (12.8–20.5)	46	20.6 (14.7–26.6)	77	37.9 (29.4–46.3)	123	28.9 (23.8–34)
2018	437,605	31	13.6 (8.8–18.3)	28	13.4 (8.4–18.4)	59	13.5 (10–16.9)	64	28 (21.1–34.8)	85	40.7 (32–49.3)	149	34 (28.6–39.5)
2019	450,645	13	5.5 (2.5–8.5)	39	18.1 (12.4–23.8)	52	11.5 (8.4–14.7)	52	22.1 (16.1–28.1)	80	37.2 (29–45.3)	132	29.3 (24.3–34.3)
2020	461,960	18	7.5 (4–10.9)	28	12.7 (8–17.4)	46	10 (7.1–12.8)	59	24.5 (18.2–30.7)	92	41.7 (33.2–50.2)	151	32.7 (27.5–37.9)
2021	450,645	21	8.6 (4.9–12.2)	34	15.2 (10.1–20.2)	55	12.2 (9–15.4)	46	18.7 (13.3–24.1)	124	55.3 (45.5–65)	170	37.7 (32.1–43.4)

^1^ Midland population age standardized to NZ Māori > 35 years and other ethnicity > 45 years. ^2^ Female and male incidences calculated by sex and age adjustment.

## Data Availability

Due to privacy, data sets generated and analyzed during the study are not publicly available.

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
