# Peer review of "The Incidence of Chronic Limb-Threatening Ischemia in the Midland Region of New Zealand over a 12-Year Period"

_jcm, 2022, doi:10.3390/jcm11123303_

Round 1

Reviewer 1 Report

The authors of the manuscript entitled "The incidence of chronic limb threatening ischemia in Midland 2 region of New Zealand over a 12-year period" aimed to define the incidence of surgical and percutaneous treated short distance IC and CLTI, and report the outcomes within the Midland region of NZ.they found that elderly age was associated with a worse survival for both short distance IC and CLTI. There was a worse survival for females with CLTI. In conclusion, PAD imposes a significant burden in NZ and further research is required to reduce this disparity.

In general this article is well written and I do not have major concerns. I do not understand two sentences (line 241-245). There is some confusion with the definitions and endpoints, which are a bit blurry, including the exclusion criteria, which could be simplified as the average angiologist may get lost after reading the methodology about which patients we are examining in this analysis.

Reviewer 2 Report

Hart et al. present an interesting study on incidence rates of PAD in New Zealand. I think the study is of interest for the reader, as it helps to complete the insights of the distribution of PAD rates around the world. However, I have a few comments: 

Materials and Methods: 

-       Is there data how the Midland population is comparable to the entire NZ population? 

-       Was bypass revision operation included, meaning were patients coming again for re-therapy included only once or multiple times? 

-       Line 86: Maybe it would be better to reword surgically treated, maybe you mean invasively treated? As you include surgically as well as endovascularly treated patients 

-       Are there only two vascular centers in the Midland region who treat those patients? Otherwise, it would be difficult to get insights of incidence rates of that region.

-       Line 111: Short distance IC should be defined more clearly, for example by adding a walking distance 

-       Line 119: How did you handle residence from the Midland region who were treated in other regions of NZ?

Results:

-       I think the results section would profit if a comparison between endo and open treatment could be added

-       Again: Survival: Is it possible to investigate in your data if there is a difference in survival whether the patient was treated endovascular vs open surgical?

Discussion:

-       Line 236: Additionally, you only included short distance IC
